# A Narrative Review of Antibiotic Prescribing Practices in Primary Care Settings in South Africa and Potential Ways Forward to Reduce Antimicrobial Resistance

**DOI:** 10.3390/antibiotics12101540

**Published:** 2023-10-14

**Authors:** Audrey Chigome, Nishana Ramdas, Phumzile Skosana, Aislinn Cook, Natalie Schellack, Stephen Campbell, Giulia Lorenzetti, Zikria Saleem, Brian Godman, Johanna C. Meyer

**Affiliations:** 1Department of Public Health Pharmacy and Management, School of Pharmacy, Sefako Makgatho Health Sciences University, Ga-Rankuwa 0208, South Africa; 210292307@swave.smu.ac.za (N.R.); stephen.campbell@smu.ac.za (S.C.); hannelie.meyer@smu.ac.za (J.C.M.); 2Department of Clinical Pharmacy, School of Pharmacy, Sefako Makgatho Health Sciences University, Molotlegi Street, Ga-Rankuwa, Pretoria 0208, South Africa; phumzile.skosana@smu.ac.za; 3Centre for Neonatal and Paediatric Infection, Institute of Infection and Immunity, St. George’s University of London, London SW17 0RE, UK; aicook@sgul.ac.uk (A.C.); glorenze@sgul.ac.uk (G.L.); 4Health Economics Research Centre, Nuffield Department of Population Health, University of Oxford, Oxford OX1 2JD, UK; 5Department of Pharmacology, Faculty of Health Sciences, University of Pretoria, Pretoria 0084, South Africa; natalie.schellack@up.ac.za; 6Centre for Epidemiology and Public Health, School of Health Sciences, University of Manchester, Manchester M13 9PL, UK; 7Department of Pharmacy Practice, Faculty of Pharmacy, Bahauddin Zakariya University, Multan 60800, Pakistan; zikria@bzu.edu.pk; 8Strathclyde Institute of Pharmacy and Biomedical Sciences, University of Strathclyde, Glasgow G4 0RE, UK; 9South African Vaccination and Immunisation Centre, Sefako Makgatho Health Sciences University, Molotlegi Street, Ga-Rankuwa, Pretoria 0208, South Africa

**Keywords:** antibiotics, antimicrobial stewardship programs, antimicrobial resistance, quality indicators, primary care, South Africa, treatment guidelines

## Abstract

There are concerns with the current prescribing of antibiotics in both the private and public primary care settings in South Africa. These concerns need to be addressed going forward to reduce rising antimicrobial resistance (AMR) rates in South Africa. Concerns include adherence to current prescribing guidelines. Consequently, there is a need to comprehensively summarise current antibiotic utilization patterns from published studies as well as potential activities to improve prescribing, including indicators and antimicrobial stewardship programs (ASPs). Published studies showed that there was an appreciable prescribing of antibiotics for patients with acute respiratory infections, i.e., 52.9% to 78% or more across the sectors. However, this was not universal, with appreciable adherence to prescribing guidelines in community health centres. Encouragingly, the majority of antibiotics prescribed, albeit often inappropriately, were from the ‘Access’ group of antibiotics in the AWaRe (Access/Watch/Reserve) classification rather than ‘Watch’ antibiotics to limit AMR. Inappropriate prescribing of antibiotics in primary care is not helped by concerns with current knowledge regarding antibiotics, AMR and ASPs among prescribers and patients in primary care. This needs to be addressed going forward. However, studies have shown it is crucial for prescribers to use a language that patients understand when discussing key aspects to enhance appropriate antibiotic use. Recommended activities for the future include improved education for all groups as well as regularly monitoring prescribing against agreed-upon guidelines and indicators.

## 1. Introduction

In 2019, an estimated 1.27 million deaths globally were directly attributed to bacterial AMR, with potentially up to 4.95 million deaths associated with bacterial AMR [1]. There are also increased morbidity and appreciable costs associated with AMR [2,3,4]. Concerns that morbidity, mortality and costs associated with AMR will continue to rise unless actively addressed have resulted in a number of international, regional and national initiatives to combat AMR. Initiatives include the World Health Organization’s Global Action Plan (GAP) to reduce AMR with the subsequent encouragement and implementation of National Action Plans (NAPs) [5,6,7]. There are also initiatives by the OECD and World Bank to suggest strategies to reduce AMR, given the human and economic consequences [3,8]. The NAPs build on the GAP with countries, including sub-Saharan Africa, at different stages of their development, implementation and monitoring [7,9,10]. These different stages of development, implementation and monitoring are influenced by a range of challenges. Challenges include available resources, which consist of both personnel and finances to implement agreed-upon activities within the NAPs [10,11]. Sub-Saharan African countries are experiencing a number of these challenges [11,12]. This is a concern as key elements of NAPs include the documentation of current antimicrobial utilization patterns across sectors as well as proposed programs to improve future antibiotic utilization and reduce AMR [6,12]. Potential programs within NAPs to decrease AMR also include developing, instigating and monitoring the implementation of agreed-upon guidelines across sectors [6,12].

South Africa appears further ahead with initiating and implementing its NAP compared with a number of other African countries [12]. Alongside this, there are multiple ongoing activities in South Africa to improve antibiotic prescribing, especially in ambulatory care, which is encouraging. Ongoing activities are listed in Table 1.

Consequently, South Africa acts as an exemplar to other African countries [12]. However, there are concerns about rising AMR rates in South Africa [12,32], exacerbated by inappropriate prescribing and dispensing practices in ambulatory care across South Africa [12,21,22,33]. Focusing on ambulatory care is important as this sector can account for up to 90–95% of human antibiotic use, especially among low- and middle-income countries (LMICs) [34,35,36]. In addition, an appreciable proportion of antibiotics are consumed for self-limiting conditions such as acute respiratory tract infections (ARIs) where antibiotics are inappropriate [37,38].

Published studies have shown variable purchasing of antibiotics without a prescription in South Africa. There was no purchasing of antibiotics in the study of Anstey Watkins et al. (2019) or among franchised pharmacies reported by Mokwele et al. (2022) [39,40]. However, purchasing antibiotics without a prescription occurred in 80% of privately owned pharmacies in the study of Mokwele et al. (2022) [40]. The purchasing of antibiotics without a prescription is also common in other African countries [41,42,43]. Similar to the situation in South Africa, variable rates of purchasing antibiotics without a prescription were seen within African countries [41]. Principal reasons for the variations seen included changing economic circumstances in the country, changes in the extent of monitoring of pharmacists’ dispensing behaviour, the location, i.e., rural vs. urban, pharmacists’ education and the infectious disease in question [41,44,45].

As mentioned, there have been concerns with the appropriateness of antibiotic prescribing in ambulatory care in both the public and private healthcare sectors in South Africa, including for ARIs [21,33,38,46]. This needs to be addressed going forward to reduce AMR in South Africa. Within ambulatory care in South Africa, the public sector is particularly important as this sector currently accounts for approximately 80% of the population [47]. However, the private sector still includes a considerable number of patients. There have also been differences in prescribing practices across sectors in other countries. For instance, in Iran, there was greater prescribing of antibiotics and injectables among the same physicians seeing private versus public sector patients [48]. In Botswana, a considerable number of patients seen in the private sector have URTIs that are subsequently treated with antibiotics [49]. This contrasts with the public system, where, alongside patients presenting with coughs, there was an appreciable number of patients presenting with vaginal discharges and sexually transmitted infections [50]. As a result, there was appreciable prescribing of combination antibiotics, including metronidazole, in the public sector, unlike the patterns seen in the private sector in Botswana [49,50].

Primary care settings in the public care sector in South Africa currently include over 3500 primary healthcare centres (PHCs) and community health centres (CHCs). CHCs and PHCs should be available within 5 km of the residency of over 90% of citizens in South Africa, as well as be free of charge to patients at the point of use [22,51]. PHCs are generally smaller than CHCs, with patients typically seen by nurses rather than physicians [52,53]. In contrast, CHCs are larger than PHCs, and they are the most visited healthcare facilities among patients in ambulatory care in South Africa. The main function of CHCs is to deliver most ambulatory care services to the citizens of South Africa. Services include advice on hygiene, vaccinations and health education, as well as antenatal care. In addition, physicians were more likely to be present in CHCs than PHCs when performing examinations and treating as well as referring patients [54,55].

It is important to treat patients with infectious diseases appropriately in primary healthcare settings not only to improve their outcomes and reduce AMR but also to conserve costs [4]. Janssen et al. (2020) recently calculated that healthcare costs associated with the inappropriate use of antibiotics to manage patients with URTIs in Ghana amounted to approximately USD 20 million annually, excluding travel costs and lost income [56]. The authors also calculated that possible savings from the appropriate management of patients with URTIs were up to USD 12 million annually [56].

Identified barriers to enhancing the appropriateness of prescribing antibiotics in primary care settings in South Africa include the pressure patients place on healthcare professionals (HCPs) to prescribe antibiotics, along with diagnostic uncertainty [57,58,59,60,61,62]. The pressures on HCPs are exacerbated in public sector primary care settings by long waiting times to see an HCP as well as limited consultation time when patients are actually seen by an HCP to discuss why they will not be prescribed an antibiotic [59]. Alongside this, HCPs themselves feel perceived rather than actual pressure from patients to prescribe antibiotics for acute infections, including ARIs, further complicating the situation as a result [38]. Excessive prescribing of antibiotics in primary healthcare settings is also seen in a number of African countries, e.g., in Ghana, where up to 86% of patients attending both private and public PHCs with upper respiratory tract infections (URTIs) were prescribed antibiotics, which were mostly broad-spectrum antibiotics [63]. Alongside this, there has been poor adherence to current guidelines when treating patients with community-acquired pneumonia in ambulatory care in Ghana [64]. In Kenya, ARIs were a common diagnosis among patients attending outreach clinics [65]. However, antibiotics were prescribed in 78.5% of patients, exacerbated by high patient loads and clinicians perceiving they should prescribe antibiotics due to a lack of access to laboratory tests [65]. There were also high rates of antibiotic prescribing for patients with URTIs (78%) attending public healthcare clinics in Namibia [66]. However, this is not universal in Namibia, with Niaz et al. (2020) demonstrating good compliance (73% of prescriptions) with current treatment guidelines [67]. There were also high prescribing rates (72.9%) for antibiotics among patients presenting with URTIs to private GPs in Botswana [49]. However, this is not always the case in South Africa. High adherence rates to infectious disease treatment guidelines have been seen among HCPs in CHCs [55], as well as HCPs in PHCs in studies conducted by Sooruth et al. (2015) and van Hecke et al. (2019) [68,69]. 

Antimicrobial stewardship programs (ASPs) have been successfully introduced in LMICs, including African countries, to improve future antimicrobial prescribing [70,71,72,73]. This was despite concerns about available trained personnel and resources in LMICs [74]. These ASPs typically include prescribing or quality indicators, with adherence to current guidelines increasingly seen as good quality care versus the traditional criteria of the WHO and International Network for Rational Use of Drugs (INRUD) in ambulatory care [68,70,75,76]. 

We are aware that a number of ASPs have been successfully introduced in hospitals across South Africa with ongoing initiatives to further improve the situation [25,26,77,78,79,80,81]. This is important as hospital ASPs help to improve the knowledge and implementation of ASPs among HCPs working in primary care settings by acting as exemplars [82]. However, currently, less is known about ASPs being successfully introduced in primary care settings across South Africa to improve future antibiotic prescribing. This may reflect challenges with undertaking ASPs among CHCs and PHCs in South Africa [15,24]. Challenges include the ability to actively disseminate guidelines and subsequently monitor prescribing against agreed-upon guidance, especially among complex cases using paper-based data collection forms as opposed to electronic monitoring with real-time feedback, as seen in Stockholm, Sweden [83,84]. In addition, there are challenges with the training of HCPs and the purchasing of any necessary equipment among PHCs in South Africa to improve care [85,86]. Engler et al. (2021), in their recent paper, also documented challenges in undertaking ASPs in CHCs due to issues of diagnostic facilities and lack of surveillance activities, with considerable concerns with continuing the education of HCPs regarding antibiotics and ASPs [24]. These challenges need to be addressed going forward, alongside efforts to improve documentation and knowledge of allergies to specific antibiotics, including penicillins [87,88,89], to enhance future appropriate antibiotic use. 

It is against this background that we identified the need for and subsequent aims of this paper. In the first instance, the current evidence base regarding antibiotic prescribing in primary care settings across South Africa will be consolidated, incorporating both the public and private sectors. This also includes the rationale for any prescribing patterns seen, as well as the use of any prescribing indicators. Subsequently, the findings will be used as a starting point for initiating future research activities. In addition, suggestions are made to improve future antibiotic prescribing in both public and private primary care settings in South Africa to reduce AMR. This is the ultimate aim of this paper.

## 2. Results

The findings from the narrative review, coupled with suggested next steps for all key stakeholder groups, will be divided into sections in line with the methodology and objectives of the study. These sections include the following:Antibiotic prescribing patterns in primary care settings in both the public and private healthcare systems in South Africa;Current knowledge and attitudes regarding antibiotics, AMR and ASPs among key stakeholder groups involved in primary care in South Africa;Quality indicators that have been used in primary care settings in South Africa in recent years to improve prescribing, which can be used in future ASPs;ASPs that have been implemented in primary care settings in South Africa and beyond and their impact to act as exemplars going forward;Potential activities that can be undertaken by all key stakeholder groups in the short to medium term in South Africa to improve the appropriateness of antibiotic prescribing in the various primary care settings, thus helping to reduce AMR in the future.

### 2.1. Prescribing of Antibiotics among Public Sector Primary Care Facilities in South Africa

There have been concerns with the quality of antibiotic prescribing among public primary care settings across South Africa; however, this is not universal. Table 2 summarises key findings among the published papers discussing prescribing practices in public primary care settings, which includes both CHCs and PHCs in South Africa.

High rates of prescribing antibiotics for essentially viral infections, including ARIs, have been seen in a number of studies, exacerbated by pressure from parents or guardians. These include the findings of Truter and Knoesen (2018), where 81.3% of community pharmacists surveyed felt that antibiotics were being over-prescribed [60]. In their various publications, Blaauw and Lagarde found that 78% of simulated patients seen in public sector clinics were recommended antibiotics for acute bronchitis even though antibiotics were not clinically indicated [38,62,90]. In their study, Mathibe et al. (2020) found that 76% of children with URTIs were prescribed antibiotics for URTIs even though three-quarters of parents/guardians were not making requests for an antibiotic [91]. There was also appreciable prescribing of antibiotics for patients with UTIs in the study of Keuler et al. (2022), which included all males and 98.5% of females [92].

However, encouragingly, where documented, the majority of antibiotics prescribed, albeit often inappropriately, were typically from the ‘Access’ list rather than the ‘Watch’ list, with little or no prescribing of ‘Reserve’ antibiotics (Table 2). This is important to help reduce AMR [93,94].

Having said this, there was variable compliance with current treatment guidelines. This was sub-optimal (45.1%) in the study of Gasson et al. (2018) [21]. However, there was greater compliance among HCPs at 59.7% in the study by Govender et al. (2021) [95]. These rates, though, compromised only 41.8% of nurses having access to the latest STGs/EML in the study by Govender et al. [95]. These findings compare to the study of Skosana et al. (2022) among CHCs, where there was a 93.4% compliance rate with the current South African STG/EML for the prescribing of antimicrobials [55].

**Table 2 antibiotics-12-01540-t002:** Summary of antibiotic prescribing among public sector primary care settings across South Africa.

Author, Year and Setting	Objectives and Methodology	Summary of Key Findings Including Prescribing of Antibiotics by the AWaRe ** Classification Where Documented
Gasson et al., 2018 [21]. CHCs and community day centres	Retrospective review of antibiotic prescribing among 8 primary care facilities in the Cape Town Metro District. One CHC and one community day centre were selected at random from 8 subdistricts A total of 654 patient records were reviewed with prescribing compared with current national guidelines Reasons ascertained when prescriptions were not adherent to current guidelines	68.7% of patients had been prescribed an antibiotic Overall guideline adherence was 45.1% Adherence differed significantly between surveyed facilities and whether the prescription was for an adult or a child Principal reasons for non-adherence to guidelines included an undocumented diagnosis (30.5% of prescriptions), an antibiotic was not required (21.6%), incorrect dose (12.9%), incorrect treatment (1.5%) and incorrect duration of therapy (9.5%)
Truter and Knoesen, 2018 [60]. Community pharmacists	Determine current antibiotic prescribing habits in their locality alongside the potential rationale for prescribing practices Self-designed questionnaire among 16 community pharmacists in Nelson Mandela Bay, Eastern Cape province	81.3% of community pharmacists felt that antibiotics were over-prescribed, including for viral infections, exacerbated by pressure from patients Amoxicillin (**A**)/co-amoxiclav (**A**) were the most dispensed antibiotics in pharmacies, followed by clarithromycin (**W**), ciprofloxacin (**W**) and azithromycin (**W**) The most common diagnoses for antibiotics were URTIs and sinusitis
Wong et al., 2018 [96]. Cross-sectional survey among patients	Assess healthcare-seeking behaviours for common infectious syndromes among patients attending PHCs and community pharmacies in two townships Cross-sectional survey among residents in townships: 1442 households in Klerksdorp (North West province) and 973 households in Soweto (Gauteng province)	PHCs were the most commonly visited facility for infectious diseases HCPs in PHCs were consulted most frequently for pneumonia, influenza-like symptoms and diarrhoea in children < 5 years old
Manderson 2020 [59]. Interviews with patients or guardians	The objective was to explore providers’ and patients’ expectations for treating ARIs with antibiotics to provide future guidance Qualitative study involving observing patient/provider consultations and 65 in-depth interviews with patients, parents or guardians presenting with children among CHCs and private GPs across South Africa, coupled with interviews with health providers (26) and key informants (12)	Prescribers’ treatment decisions were typically informed by a clinical assessment, concern about the risks of bacterial infection and their perceptions of the patient’s ability to seek further care Any prescribing of antibiotics typically reflected clinicians’ appreciation of the economic constraint and the vulnerability of their patients Patients did not always request antibiotics and were willing to discuss how to manage acute conditions without them, which is encouraging for the future
Mathibe et al., 2020 [91]. Parents/guardians accompanying children to a CHC	The objective was to investigate whether parents/guardians accompanying children aged 5 years or below with URTIs treated at a CHC expected or influenced the prescribing of antibiotics Questionnaire-based study among 306 parents/guardians in Pietermaritzburg, KwaZulu-Natal province	76% of parents/guardians received antibiotics for URTIs for their children, with 67% of parents/guardians not making requests for antimicrobial therapy Overall, there was a statistically significant (*p* < 0.0001) chance of children being prescribed antibiotics for URTIs without a request for one Prescribers need education on the rational prescribing of antimicrobials and to implement evidence-based STGs to reduce inappropriate prescribing of antibiotics in children with URTIs in South Africa
Sharma et al., 2020 [97]. PHC facilities	Analysis of antibiotic consumption and procurement data in a PHC in the Eastern Cape province to drive evidence-based policies and practices ‘ABC**’ analysis of procured antibiotics	Antibiotics made up approximately 7% of the total annual pharmaceutical expenditure in this PHC facility The most procured antimicrobials were (1) isoniazid; (2) flucloxacillin (**A**); (3) azithromycin (**W**); (4) a combination of rifampicin, isoniazid, pyrazinamide and ethambutol; and (5) amoxicillin (**A**) Totals of 55%, 2% and 15% of antibiotics accounted for ‘Access’, ‘Watch’ and ‘Access and Watch’ categories, respectively, with the remaining 28% of antimicrobials for antituberculosis medicines No ‘Reserve’ antibiotics were procured
Govender et al., 2021 * [95]. PHC facilities	The aim was to evaluate the implementation and utilization of STGs/EML by prescribers (98 nurses) in PHCs In addition, another aim was to determine their knowledge of the use and implementation of guidelines and review their adherence to guidelines when prescribing medicines An observational quantitative descriptive research design was used	Only 41.8% of nurses had access to the latest STGs/EML A total of 78.6% of nurses used and 21.4% sometimes used the STGs/EML often when making prescribing decisions A total of 78.3% of prescriptions had the diagnosis recorded, with 59.7% of prescriptions adherent to the STG/EML recommendations A total of 94.9% of nurses stated they wished to receive training on the use of the STGs/EML
Keuler et al., 2022 [92]. PHCs	A retrospective multicentre medical records review of patients with urinary tract infections (UTIs) was conducted in primary care facilities of the Western Cape public sector for the period 1 October 2020–28 February 2021 The objectives were to describe how UTIs are treated in PHC and to determine compliance with current local guidelines (STGs/EML)	A total of 401 UTI episodes from 383 patients were reviewed from six PHCs Antibiotics were prescribed in all male and 98.5% of female episodes as well as in all uncomplicated episodes and 98.3% of complicated episodes The allergy status of patients was recorded in 88.3% of patient folders Nitrofurantoin (**A**) was prescribed in the majority of UTI episodes (57.1%), followed by ciprofloxacin (39.7%) (**W**) Nitrofurantoin was more frequently prescribed in patients with uncomplicated (75.0%) versus complicated (50.8%) UTI episodes and in episodes in women (63.9%). In terms of compliance with STGs, nitrofurantoin was appropriately selected in 75.0% of uncomplicated episodes. However, for complicated UTIs, compliance in terms of antibiotic choice was better for ciprofloxacin (44.4%) than for nitrofurantoin (25.6%). Overall compliance in prescribing was greater for uncomplicated UTIs (61.5%) than for complicated UTIs (52.9%) Failure to comply with STG recommendations was mostly due to inappropriate antibiotic selection for complicated UTIs and duration of therapy
Skosana et al., 2022 [55]. CHCs	The aim of the study was to assess current antibiotic prescribing practices among 18 CHCs throughout South Africa, including adherence to current guidelines The methodology involved a point prevalence survey design among CHCs across South Africa	The prevalence of antimicrobial use among patients attending the CHCs was 21.5% (420/1958 patients), including one or more antimicrobials per patient The most frequently prescribed antimicrobials were amoxicillin (32.9%) (**A**); isoniazid (11.3%); and a combination of rifampicin, isoniazid, pyrazinamide and ethambutol The most common indication for antibiotics were ear, nose and throat infections (22.8%), with no culture results recorded ‘Access’ antibiotics accounted for 80.5% of antibiotics prescribed, with none from the ‘Reserve’ list There was 93.4% compliance with the current South African STG/EML for antimicrobials
Lagarde and Blaauw, 2019 and 2023 [38,62,90]. Simulated patients	The objective was to assess prescribing practices for young and healthy simulated patients (SP) presenting with viral bronchitis in the city of Johannesburg to both private and public PHCs A total of 102 SP visits in the public sector (and 99 in the private sector) One hundred twenty-five providers (across sectors) were interviewed face-to-face	Antibiotics were recommended in 72.6% of consultations, higher in the public sector (78.4%) versus the private sector (66.7%), enhanced by perceived patient pressure to prescribe antibiotics for this condition 91% of patients who received antibiotics in the public sector were prescribed amoxicillin (**A**) Antibiotics from the ‘Watch’ list were prescribed in only 5% of consultations in the public sector (versus 20% in the private sector) 84% of providers knew that the SP case was likely a viral infection (77% in the public sector vs. 88% in the private sector) 58% of providers believed antibiotics would not hasten recovery (40% public vs. 68% private; *p* = 0.002) 47% of public providers thought patients would not come back if antibiotics were not prescribed (47% vs. 72% in the private sector; *p* = 0.008) Lower awareness of AMR among public providers compared with private ones (*p* < 0.001)

NB: * refers to STGs/EML for all indications; ** ABC: ’A’ refers to the most important medicines in the facility, ‘B’ less important and ‘C’ the least important [97]; ARIs: Acute Respiratory Infections; AWaRe classification: Access (**A**) and Watch (**W**) antibiotics [34,93,94]; CHCs: Community Healthcare Centres; GP: General Practitioner; HCP: Healthcare Professional; PHCs: Primary Healthcare Clinics; STGs/EML: Standard Treatment Guidelines/Essential Medicine List; URTIs: Upper Respiratory Tract Infections; UTI: Urinary Tract Infection.

### 2.2. Prescribing of Antibiotics among Private General Practitioners (GPs) in South Africa

There have also been concerns about the quality of antibiotic prescriptions among private GPs across South Africa. Table 3 provides a summary of current prescribing practices among private GPs, especially for ARIs.

There were concerns with antibiotic prescribing among private GPs in their various publications of Blaauw and Lagarde. The authors showed that antibiotics were being recommended in two-thirds of simulated patients presenting to private GPs with acute bronchitis even though antibiotics were typically not clinically indicated [38,62,90]. Guma et al. (2022) also found high rates of inappropriate antibiotic prescribing. The authors found that 55.5% of surveyed private GPs prescribed antibiotics empirically for patients with ARIs more than 70% of the time [33]. Ncube et al. (2017) similarly found that 52.9% of patients in medical aid schemes (private insurance) with acute bronchitis were prescribed an antibiotic when consulting a medical practitioner [98]. Manderson (2020) also found that GPs in private practice in South Africa often provided a prescription for patients with ARIs; however, this was sometimes post-dated to help discourage antibiotic use [59].

However, encouragingly, where documented (Table 3), the majority of antibiotics prescribed, albeit often inappropriately, were typically from the ‘Access’ list rather than the ‘Watch’ list, with little or no prescribing of ‘Reserve’ antibiotics. This is similar to the situation with HCPs in public primary healthcare settings (Table 2).

### 2.3. Knowledge and Attitudes Regarding Antibiotics and AMR among Key Stakeholder Groups Involved in Primary Healthcare in South Africa

There are concerns with the current knowledge regarding antibiotics, AMR and ASPs among all key stakeholder groups in South Africa. This principally includes HCP prescribers, patients and healthcare students (Table 4). These concerns need to be addressed going forward to help reduce AMR. 

Issues and concerns identified among the 11 studies in this narrative review included concerns regarding antibiotics, AMR and ASPs. Alongside this, an appreciable number of final-year medical students did not feel confident to prescribe antibiotics post qualification [29]. In addition, both medical and pharmacy students wanted more education in the curricula on key aspects of antibiotics, AMR and AMS, to assist with treatment decisions post qualification [29,101]. There are also concerns with the knowledge of HCPs in practice regarding these key issues [38,57].

The published studies (Table 4) have also shown that it is crucial that HCPs use a language that patients will understand when discussing key aspects of antibiotics and AMR with them; otherwise, critical information will not be fully understood by patients. 

Overall, future activities include upgrading the curricula for HCPs to better equip them with the necessary confidence and skills regarding appropriate antibiotic prescribing across sectors [102]. In addition, it is necessary to better equip HCPs for meaningful communication with patients on key aspects regarding antibiotics and AMR during consultations. Universities also need to upgrade their continuous professional development (CPD) activities, given concerns with the knowledge of HCPs in practice in primary care settings across South Africa.

**Table 4 antibiotics-12-01540-t004:** Knowledge, attitudes and perceptions regarding antibiotics, AMR and ASPs among key stakeholder groups in South Africa.

Author, Year and Setting	Objectives and Methodology	Key Findings
Burger et al., 2016 [101]. Questionnaire study at 8 universities training pharmacists	Ascertain pharmacy students’ perceptions, attitudes and knowledge regarding antibiotic use and AMR and their perceived quality of education relating to antibiotics and infection Study was undertaken using a self-administered questionnaire among 978 students	The majority of respondents were familiar with ASPs in South Africa The most important interventions to combat AMR are considered to be education on antimicrobial therapy, the development of new antimicrobials and the development of institutional guidelines for antimicrobial use There were significant differences in the responses between the universities pertaining to the role of the pharmacist in ASPs as well as the number of hours spent teaching about AMS at the different universities The majority of the respondents wanted more training on AMS at their university
Wasserman et al., 2017 [29]. Self-administered questionnaire among final-year medical students at 3 universities in South Africa	Ascertain medical students’ perceptions, attitudes and knowledge regarding antibiotic use and AMR and their perceived quality of education relating to antibiotics and infection Self-administered questionnaire among 289 medical students	92% of participating medical students agreed that antibiotics are overused, and 87% agreed that AMR is a significant problem in South Africa 95% of students reported they would appreciate more education on appropriate use of antibiotics, with only 33% currently feeling confident enough to prescribe antibiotics Prescribing confidence was associated with the use of antibiotic STGs, familiarity with AMS and more frequent contact with infectious disease specialists Medical students who had used antibiotic STGs and found their education more useful scored higher on the knowledge questions
Farley et al., 2018 [57]. Cross-sectional survey among PHC prescribers	Ascertain knowledge, attitudes and perceptions towards AMR among 264 PHC prescribers Questionnaire survey among prescribers, principally from the private sector	95.8% believed that AMR is a significant problem in South Africa; however, 66.5% felt pressure from patients to prescribe antibiotics Median knowledge score was 5/7, highest in prescribers aged < 55 years (*p* = 0.0001) Prescribers with higher knowledge scores were more likely to believe that prescribing narrow-spectrum antibiotics where necessary would reduce AMR as opposed to prescribing broad-spectrum antibiotics. In addition, they were more likely to report that explaining disease features to patients was a useful alternative to prescribing Overall, additional educational input is needed to enhance appropriate antibiotic prescribing among prescribers in primary healthcare settings in South Africa
Anstey Watkins et al., 2019 [39]. Semi-structured interviews with 60 rural village residents	Ascertain key issues regarding antibiotics—including knowledge of antibiotics and AMR—among 60 village residents in rural South Africa	Most respondents used the government public health system for treatment Most interviewees had not heard of the term ‘antibiotics’ before; however, when described using relevant Xitsonga words, most interviewees could relate to this term A number of interviewees admitted not finishing their prescribed course and either disposed of unused antibiotics into outdoor latrines or used them for the next infection The term AMR was unfamiliar to all interviewees
Farley et al., 2019 [103]. Cross-sectional survey of patients attending both public and private primary healthcare facilities	Using a cross-sectional survey to assess knowledge, attitudes and perceptions concerning key aspects of antibiotics among 782 patients attending private and public healthcare facilities across South Africa	61% of surveyed patients believed that antibiotics would work less well in the future if overused now 82% of patients believed that antibiotics are a strong form of medication and should only be taken when absolutely necessary, with 53% stating they get worried when antibiotics are prescribed as they prefer not to take them 72% of patients believed it was the human body that becomes resistant to antibiotics, 66% believed that antibiotics are good for treating viruses, and 25% of patients believed that patients should be given antibiotics on demand when visiting HCPs Overall, patients with high knowledge scores exhibited more protective behaviours and beliefs towards antibiotics, with mean knowledge scores lower among public vs. private sector surveyed patients
van Hecke et al., 2019 [69]. Interviews with HCPs	Qualitative semi-structured interviews among 23 HCPs in PHCs regarding antibiotic prescribing decisions for 2 common infections—acute cough and UTIs	Communication difficulties between HCPs and patients often hampered efforts to explain a non-antibiotic management strategy for the management of ARIs The local African languages and associated relative paucity of scientific medical terms currently hamper patient education regarding the differences between bacterial and viral infections and hinders efforts to discuss more complex issues, including AMR. This needs addressing going forward
Manderson, 2020 [59]. Interviews with patients or guardians	Qualitative interviews among prescribers or patients/their guardians to explore providers’ and patients’ expectations for treating ARIs with antibiotics	Differences in how prescribers and patients understood AMR with patients often perceiving AMR in reference to immunity, ineffectiveness of an antibiotic or occasionally an allergy, e.g., a patient might develop resistance to penicillin, i.e., a negative reaction. Instead, require a different antibiotic or they might develop resistance to a particular antibiotic because it is ineffective for them even if effective for other patients. Alternatively, the bacteria might be considered resistant to the antibiotic because the choice of therapy was incorrect Some prescribers mentioned that bacteria develop AMR genetically to an antibiotic associated with inappropriate prescribing More often, prescribers, as well as patients, saw AMR as a problem at the patient level
Balliram et al., 2021 [30]. Cross-sectional survey among HCPs using a self-administered questionnaire	Assess knowledge, attitudes and practices on antimicrobials, AMR and AMS among HCPs nationally Questionnaire-based survey amongst 1120 doctors, 744 pharmacists and 659 nurses	AMR was considered a severe problem nationally and globally by the majority of participants, with knowledge gaps in their understanding of antimicrobials, AMR and AMS Confidence scores with prescribing were highest among doctors (57.8%), nurses (45.3%) and pharmacists (32.9%); however, little or no confidence in using combination therapy 91.2% of participants stated educational campaigns would help combat AMR, with 81.6% requesting more training and education, with only a minority (40.1%) previously attending training on these topics
Mokoena et al., 2021 [104]. Semi-structured questionnaire among taxi drivers	The objectives were to assess knowledge and understanding among the minibus-taxi community on antibiotics and AMR In addition, document antibiotic terminology used across the district A semi-structured questionnaire was administered to 83 minibus–taxi community members	71% of participants knew the importance of taking antibiotics as directed 64% believed it was acceptable to share antibiotics 75% thought AMR occurred in the human body One misconception among participants was that antibiotics treated cold/flu and fever 80% of participants had never heard of AMS, ‘antibiotic- resistant bacteria’, ‘drug resistance’ or ‘AMR’ 51% and 52% of participants, respectively, had heard of the terms ‘AMR’ and ‘superbugs’ from various sources including doctors, nurses and pharmacists
Lagarde and Blaauw, 2019 and 2023 [38,62]. Simulated patients	The objective was to assess prescribing practices for young and healthy simulated patients (SP) presenting with viral bronchitis in the city of Johannesburg A total of 99 SP visits in the private sector and 102 in the public sector One hundred twenty-five providers (across sectors) were interviewed face-to-face	A total of 84% of providers knew that the SP case was likely a viral infection (88% in the private sector vs. 77% in the public sector) A total of 58% of providers believed antibiotics would not hasten recovery (68% in the private sector vs. 40% in the public sector; *p* = 0.002) However, more than half of SPs still received antibiotics for their acute bronchitis even after stating they did not want an antibiotic—exacerbated by 72% of private providers believing patients would not come back if no antibiotic were prescribed (72% vs. 47% in the public sector; *p* = 0.008) Higher awareness of AMR among private vs. public providers (*p* < 0.001) The results suggest extensive educational activities are needed among both prescribers and patients to address entrenched prescribing habits

NB: AMR: antimicrobial resistance; AMS: antimicrobial stewardship; ARIs: acute respiratory illnesses; ASP: antimicrobial stewardship programme; HCPs: healthcare professionals; PHCs: primary healthcare clinics; STGs: Standard Treatment Guidelines; UTIs: urinary tract infections.

### 2.4. Quality Indicators Currently Being Used in Primary Healthcare in South Africa

A number of prescribing and quality indicators have been used in South Africa to assess the quality of current prescribing in primary healthcare settings across the sectors. These are contained in Table 5. Documented prescribing and quality indicators can be part of future ASPs, with, for instance, adherence to guidelines increasingly seen as providing good quality care in South Africa and beyond [75,76].

**Table 5 antibiotics-12-01540-t005:** Prescribing and quality indicators currently being used in primary care settings across South Africa.

Indicator (Activity/Performance Indicators)	Reference
% of monthly antibiotics used (defined daily doses per 100 prescriptions dispensed)	[89]
% of patients prescribed an appropriate antibiotic dose and duration for their diagnosed infectious disease	[89,105]
% of patients prescribed an antibiotic (empirically) for an ARI/URTI	[33,39,91,98]
% of adherence to a bundle of antibiotic prescribing process measures (allergies documented, diagnoses provided, appropriate prescribing according to current guidelines, appropriate doses of antibiotics prescribed, their frequency and duration, as well as a valid prescription (prescriber’s name, signature and date))	[89]
% of prescriptions adherent to current guidelines	[21,22,95]
% of appropriate prescriptions (according to current guidance)	[46]
% of antibiotics prescribed/procured broken down by AWaRe * categories	[46,55,97]

NB: ARI: acute respiratory illness; * AWaRe classification [93,94]; URTI: upper respiratory tract infection.

### 2.5. Antimicrobial Stewardship Programs in Primary Care Facilities in South Africa

A number of ASPs have already been undertaken in primary care settings in South Africa to improve future prescribing. These exemplars (Table 6) can provide direction to all key stakeholder groups in South Africa as part of proposed future activities (Table 7 and Appendix A) to address concerns with current AMR rates in the country. 

Blaauw and Lagarde (2019) showed that enhancing the knowledge of patients so that they could state they did not want antibiotics to treat acute bronchitis helped to reduce unnecessary prescribing [38]. However, more needs to be done to continue to reduce unnecessary prescribing (Table 7 and Appendix A). Alongside this, De Vries et al. (2022) showed that regular multidisciplinary audits and feedback meetings appreciably enhanced adherence to guidelines and subsequent antibiotic use [89]. However, this was more difficult during the winter months, especially for essentially viral infections. 

Table 7 provides a summary of suggested activities among all key stakeholders in the short to medium term. Further details are included in Appendix A. These activities build on current activities instigated by the South African Department of Health and others to improve antibiotic prescribing across South Africa (Table 1). 

## 3. Discussion

We believe this is the first study in South Africa to comprehensively review all aspects of antibiotic prescribing in the primary care setting in South Africa, which includes both the public and private healthcare sectors. Key aspects of the narrative review include documenting current antibiotic prescribing patterns as well as knowledge and perceptions among all key stakeholder groups towards antibiotics, AMR and ASPs. In addition, pertinent key activities to improve future prescribing. The latter includes developing pertinent prescribing or quality indicators, introducing ASPs and subsequently monitoring their impact.

The findings highlight that there can be concerns with the current prescribing of antibiotics in primary healthcare settings across the sectors in South Africa [21,22,33,38,60,62,91,98]; however, this is not always the case [55,59] (Table 2 and Table 3). These concerns with the prescribing of antibiotics in South Africa are similar to those of a number of other African countries. Excessive and inappropriate prescribing of antibiotics is often seen in primary care among other African countries [49,64,108,109,110,111], with associated cost and adverse reaction implications alongside increasing AMR [56,99,112,113,114]. Encouragingly, the majority of antibiotics prescribed across both sectors in South Africa were typically from the ‘Access’ list rather than the ‘Watch’ list, with little or no prescribing of ‘Reserve’ antibiotics where the nature and content of prescriptions were documented (Table 2 and Table 3). This is an important first step to reduce AMR as we have seen high rates of ‘Watch’ antibiotics being prescribed and dispensed in ambulatory care in other LMICs, which needs to be avoided where possible [115,116,117,118].

Key activities to improve future prescribing of antibiotics in primary care in South Africa include improving the evidence base as well as seeking to introduce easy-to-use Apps or other approaches that enhance routine data collection in electronic formats (Table 7). Such approaches are essential for the Ministry of Health and health insurers/medical aid societies to be able to rapidly monitor the appropriateness of any antibiotics prescribed (Table 7). These are ongoing projects in South Africa.

There are also concerns with current sub-optimal adherence to guidelines among a number of prescribers in both the primary care sectors in South Africa [21,22,46,62,92,107], which is also similar to other African countries [65,119,120,121,122]. However, this is not universal, as recently seen, for example, among public CHCs across South Africa in the study of Skosana et al. (2022) [55]. The development of pertinent prescribing and quality indicators, including prescribing against AWaRe guidance, are important going forward alongside measures to improve adherence to guidelines. These combined measures should help enhance appropriate antibiotic prescribing in primary healthcare in South Africa, given current concerns (Table 2 and Table 3). Alongside this, there is a need for improved education of all key stakeholders regarding antibiotics, AMS, ASPs and AMR (Table 7 and Appendix A). 

A number of activity or process indicators have already been used in South Africa to improve antibiotic prescribing in ambulatory care (Table 5). However, these need to be refined, especially with the recent availability of the AWaRe book which gives universally accepted first- and second-line treatment guidance [34,123]. Any agreed-upon prescribing or quality indicators can subsequently be incorporated into future ASPs to reduce AMR in line with the goals of the NAP [12,16]. However, for such activities to be effective and achieve target goals, a number of co-ordinated activities and technologies need to be in place. In a number of countries and settings, we have seen that disjointed activities, i.e., those not involving all key stakeholders in the reforms, fail to achieve target objectives [41,124,125,126]. Consequently, it is essential that key groups work together in a co-ordinated fashion and that HCP prescribing habits are regularly monitored, with the findings regularly and rapidly fed back to prescribers. This has worked well in Stockholm, Sweden, with the development of an evidence-based ‘Wise list’ of medicines [83,84]. The list is evidence-based and comprehensively communicated to physicians and patients, with physicians’ prescribing patterns against agreed-upon guidance regularly fed back to them via an online system [84,127]. The well-accepted methodology for compiling the list, coupled with physician education and regular feedback, has resulted in high compliance rates to the ‘Wise list’ in practice [83,84]. With biosimilars, we have also seen that countries that have introduced multiple measures and initiatives, including education of physicians and patients, combined with prescribing targets, have seen an accelerated uptake of biosimilars [124,128]. This situation compares with the limited use of biosimilars in countries with limited demand-side measures encouraging their use [124,129]. In the future, online systems need to be in place in South Africa to routinely record and provide feedback to the HCPs working in primary care facilities on their prescribing patterns against agreed-upon guidance, thus ensuring maximum adherence to prescribing guidance in practice. Ad hoc projects, especially those using paper-based records severely impacting the ability to easily collect data, will not have the same impact in practice [37,124,130]. Technologies that can be introduced to enhance routine monitoring of primary care prescribing could include the instigation of easy-to-use Apps, thereby building on ongoing projects. 

Ongoing ASPs in Africa, including South Africa, and other LMICs (Table 6 and Appendix A) can act as exemplars for future ASPs in primary care settings in South Africa. We are aware that undertaking ASPs in primary healthcare in LMICs is challenging due to personnel and resource challenges [74]; however, as seen, this is beginning to change (Table 6 and Appendix A). This is important given the need to improve antibiotic prescribing in ambulatory care across Africa if AMR rates are to be reduced in this high-priority region [1].

Patients and their organizations’ activities have typically been a forgotten element in improving the appropriateness of antibiotic prescribing across LMICs. However, their role in influencing prescribing is increasingly being recognised, including within South Africa. However, concerns with their knowledge regarding antibiotics, AMR and ASPs need to be addressed going forward (Table 2, Table 3 and Table 4). Potential activities include more targeted research to improve the current evidence base in South Africa (Table 7 and Appendix A). Concerns with knowledge among patients include their beliefs that antibiotics can cure self-limiting conditions, including URTIs (Table 4), which is similar to other LMICs [37,38,62]. 

We have already seen with the COVID-19 pandemic that misinformation regarding the possible role of hydroxychloroquine and ivermectin can fuel inappropriate demand in South Africa [131]. Consequently, understanding the potential role of social media, as well as making sure patients are familiar with the terminology used, given previous concerns in South Africa [39,103,104], are key activities going forward in South Africa to address this challenge. As part of this, there needs to be multidisciplinary collaboration among HCPs, policymakers, patients and patient advocacy groups across the sectors to develop patient-centred educational programmes and ASPs that address current concerns (Table 7). Subsequently, the influence of patients/patient advocacy groups should be used to improve future antibiotic prescribing. In view of this, patient representatives and advocacy groups should be included in the development and implementation of any prescribing or quality indicators as part of planned ASPs (Appendix A). This is because patients/patient advocacy groups can provide valuable insights and perspectives on patient needs, expectations and concerns related to current antibiotic use. In view of this, efforts should be made to enhance communication between HCPs and patients regarding the appropriate management of self-limiting infections such as ARIs (Table 7). Clear and accessible information about non-antibiotic management strategies, treatment expectations, and the potential risks of inappropriate antibiotic use, should also be provided to patients in languages that can be easily understood. We will continue to monitor the situation, given ongoing concerns across sectors and groups in South Africa.

We are aware of a number of limitations with this paper. These include the fact that we did not undertake a full systematic review for the reasons provided. However, we have included an appreciable number of papers discussing the current situation regarding prescribing patterns of antibiotics in primary healthcare settings across South Africa, the possible rationale for the patterns seen, and potential ways to address concerns moving forward. The latter has been achieved with input from senior-level co-authors from across Africa and beyond. Despite these limitations, we believe our findings and suggestions for the future are robust.

## 4. Materials and Methods

### 4.1. Our Approach and Key Questions

The principal approach used to address the objectives of this paper was a narrative review of key questions [73,132,133,134]. With the ultimate aim of reducing AMR in South Africa in the future, the six key questions to address included the following: What have been the antibiotic prescribing patterns in public sector primary care settings across South Africa in recent years?What have been the antibiotic prescribing patterns among private GPs across South Africa in recent years?What is the current knowledge and what are the attitudes regarding antibiotics, AMR and ASPs among key stakeholder groups involved in primary care in South Africa?What prescribing and quality indicators have been used in primary care settings in South Africa to improve prescribing in recent years?What ASPs, including their impact, have been implemented in primary care settings to date across South Africa to improve future antibiotic prescribing? Similarly, what future guidance can other LMICs provide to key stakeholder groups in South Africa?What potential activities can be undertaken by all key stakeholder groups in South Africa in the short to medium term to improve the appropriateness of antibiotic prescribing among prescribers in the various primary care settings, thereby reducing AMR in South Africa in the future?

We were aware that there have been reviews of factors affecting the prescribing of antibiotics for essentially primary care settings as well as interventions to reduce inappropriate prescribing of antibiotics. However, a number of these reviews have principally focused on higher-income countries where resources and personnel to influence and monitor prescribing can be very different [73,74,133,135,136,137]. In addition, typically focused on one key area without bringing together all the key aspects associated with improving antibiotic prescribing in primary care settings into one comprehensive study.

We identified the narrative review approach as the most appropriate for the purpose of this paper since this approach allows for a broader scope compared to a systematic review. This is because a number of potential papers may not be listed in PubMed or Web of Science, although they provide useful insights regarding current practices in South Africa. Furthermore, pertinent data may well be part of a wider paper, which is likely to be missed in a systematic review. This includes assessing inappropriate prescribing of antibiotics in patients presenting with other infectious diseases, including HIV [100]. 

In addition, a narrative review allows more flexibility and broader coverage of the relevant literature in a particular field. However, we do acknowledge the limitations of this approach in terms of synthesis and rigour. To minimise selection bias and ensure that relevant information is included in the narrative review, the chosen participating co-authors have considerable experience across Africa and beyond in terms of practice and research surrounding the management and prescribing of antibiotics in ambulatory care. Alongside this, considerable experience with implementing policies to improve appropriate prescribing. This includes the development of pertinent quality indicators, implementation of ASPs and research to evaluate their implementation. 

We have adopted a similar approach in the past with good results when previously documenting and suggesting activities to improve the management of patients with non-infectious and infectious diseases across Africa, as well as challenges with implementing NAPs to reduce AMR and potential ways to address these [12,124,130,138,139,140,141,142]. Consequently, we believed this would be an appropriate approach for this comprehensive study.

### 4.2. Search Strategy and Inclusion Criteria 

A literature search was performed to address the six identified questions using a number of databases, including Google Scholar and PubMed/MEDLINE. In addition, a manual search of the grey literature was undertaken, which included key Ministry of Health documents in South Africa.

The search strategy to address the identified questions used a number of search terms. These included ambulatory care; antimicrobial prescribing; antibiotic prescribing; antimicrobial stewardship; antimicrobial stewardship programs; community healthcare centres; community day centres; guidelines; guidelines adherence; low- and middle-income countries; prescribing indicators; quality indicators; primary health facilities; primary healthcare centres; and South Africa.

Given the likely scarcity of published literature relating to these six key questions, the qualifying criteria were purposefully wide in order to maximise the sensitivity of the search. However, only English language papers were sourced, with English being the international scientific language. 

We also only concentrated on primary healthcare settings as this is where the majority of patients with infections such as URTIs are treated in South Africa [55]. As a result, we excluded studies conducted in hospital outpatients unless they were part of ASPs. In addition, we excluded any studies conducted in emergency room settings as well as any that involved more specialist PHCs, such as those in correctional centres. This is because there is likely to be a greater prevalence of STIs in correctional centres, which could potentially bias the findings [22]. 

We also only concentrated on documented studies from 2015 onwards with ongoing attempts to instigate universal healthcare (UHC) in South Africa [47]. In addition, we considered the excessive prescribing of antibiotics in patients with COVID-19 since the start of the pandemic in 2020 among LMICs, which occurred despite limited evidence of bacterial co-infections or secondary infections [143,144,145,146,147,148]. This has been exacerbated in Africa with antibiotics being included in national treatment guidelines despite COVID-19 being a viral infection [149]. 

The introduction of UHC has important implications for the provision of healthcare in the ambulatory sector in South Africa, especially considering that the private sector currently accounts for only 20% or less of the population, and the standard of care between the sectors has varied [47,62]. However, the private sector is still important since this sector has been identified as a priority in South Africa’s National Strategic Plan on Human Immunodeficiency Virus (HIV), Tuberculosis (TB) and Sexually Transmitted Infections (STIs) [99,150] and still accounts for an appreciable number of patients in South Africa [47]. 

Achieving appropriate antibiotic prescribing in primary care settings is also a critical part of achieving UHC in South Africa. Consequently, it is important that potential interventions are prioritised in ambulatory as opposed to hospital care, which has been the principal emphasis to date [95,130].

With respect to private insurance companies, we included a wide range incorporating both health insurers and medical aid groups in addition to the ongoing National Health Insurance (NHI) bill currently being implemented [33,151,152]. 

### 4.3. Documentation Strategy and Suggestions for the Future

All documented studies included the authors, publication year, a summary of the methodology and objectives, as well as key findings, and acknowledging whether the surveyed HCPs were treating private or public patients. The importance thereof is because we are aware that prescribing patterns may vary with increasing focus on prescribing habits, especially following increased monitoring of the NAP and the introduction of NHI in South Africa.

Where possible, reported antibiotic utilisation was broken down by its AWaRe classification of ‘Access’, ‘Watch’ or ‘Reserve’ [93,94]. The ‘Access’ group of antibiotics is considered first- or second-line antibiotics for common or severe clinical syndromes, typically having a narrow spectrum as well as low resistance potential. There is a higher potential for resistance and side-effects among antibiotics in the ‘Watch’ group; consequently, their prescribing should be carefully considered by healthcare professionals across the sectors. This is in line with recommendations in the AWaRe report [34,123,153]. The ‘Reserve’ group should rarely, if ever, be prescribed in ambulatory care; ideally, it is only prescribed as last resort antibiotics in hospitals [34,94,123]. The initial target for ‘Access’ antibiotics is 60% of total utilisation across sectors; however, this will vary across countries [34,154].

Finally, with respect to possible future strategies among all key stakeholder groups, as mentioned, we will build on the considerable experience of the co-authors. As also mentioned, we have used similar approaches before when documenting and suggesting activities to improve the management of patients with non-infectious and infectious diseases, as well as challenges with implementing NAPs to reduce AMR across Africa and potential ways to address this [12,130,138,139,140,141,142].

## 5. Conclusions

In conclusion, there are ongoing concerns regarding the extent of inappropriate prescribing of antibiotics in primary care in South Africa and the implications for increasing AMR. A number of activities are essential among all key stakeholder groups to address the current situation, which are contained in Table 7 and Appendix A. Potential activities include a greater evidence base regarding current prescribing patterns and possible ways to improve the monitoring of prescribing against agreed-upon indicators and guidelines. Easy-to-use Apps are a potential way forward. However, as part of the implementation of universal healthcare in South Africa, electronic health records need to be introduced across all sectors. Similarly, greater knowledge regarding the role of patients in improving future antibiotic use is also essential going forward, given their increasing influence. 

The development of pertinent prescribing or quality indicators is also an essential next step to improve future prescribing as part of planned ASPs, building on examples in other countries (Table 6 and Appendix A). A number of ASPs have already been undertaken in primary care settings in South Africa and among LMICs, which can act as exemplars going forward. Successful activities to improve future antibiotic prescribing in South Africa will typically involve comprehensive activities among all key stakeholder groups, with education as a key component. We will continue to monitor the situation given its current urgency and the need to reduce AMR in South Africa as part of ongoing NAPs.

## Figures and Tables

**Table 1 antibiotics-12-01540-t001:** Ongoing activities by the Department of Health and others in South Africa to improve antibiotic prescribing, particularly in ambulatory care.

Activity	Reference
Regular monitoring of the implementation of the National Action Plan/Antimicrobial Resistance National Strategy Framework 2017–2014 alongside active surveillance of AMR	[12,13,14,15,16]
Updating of Standard Treatment Guidelines/Essential Medicine List (STG/EML—2020), including recommendations for the management of COVID-19 and the management of urinary tract infections in primary care	[17,18,19]
Developing and broadcasting a national manual to improve infection prevention and control across sectors	[20]
Assessment and monitoring of prescribing of antibiotics in ambulatory care vs. recommendations in the STG/EML	[21,22]
Encouraging citizens to become antibiotic guardians	[23]
Assessing antimicrobial stewardship activities among public healthcare facilities in South Africa and encouraging the implementation of ASPs	[24,25,26]
Refining curricula among student healthcare professionals to improve knowledge regarding antibiotics, AMR and ASPs, as well as continuous professional development activities post qualification to address knowledge and training gaps	[27,28,29,30]
One Health approach to limit the prescribing of colistin	[31]

NB: AMR: Antimicrobial Resistance; ASP: Antimicrobial Stewardship Program.

**Table 3 antibiotics-12-01540-t003:** Summary of antibiotic prescribing among private GPs in primary care settings across South Africa.

Author, Year and Setting	Objective and Methodology	Summary of the Key Findings Including Prescribing of Antibiotics by the AWaRe * Classification Where Documented
Ncube et al., 2017 [98]. Private health insurance (medical aid) schemes	The objective was to analyse antibiotic prescription patterns among medical practitioners in private health insurance schemes treating patients with acute bronchitis Claims among members of 11 health insurance schemes treating these patients were analysed	52.9% of patients with acute bronchitis were prescribed an antibiotic Patients with viral bronchitis were more likely to be prescribed an antibiotic than those with bacterial bronchitis Patients with a chronic illness were less likely to be prescribed an antibiotic than those without a chronic illness Of the antibiotics prescribed, >70% were penicillins, cephalosporins and other beta-lactams
Truter and Knoesen, 2018 [60]. Community pharmacists	Determine current antibiotic prescribing habits in their locality alongside the potential rationale for prescribing practices Self-designed questionnaire among 16 community pharmacists in Nelson Mandela Bay, Eastern Cape province	81.3% of community pharmacists felt that antibiotics were over-prescribed, including for viral infections, exacerbated by pressure from patients Amoxicillin (**A**)/co-amoxiclav (**A**) were the most dispensed antibiotics in pharmacies, followed by clarithromycin (**W**), ciprofloxacin (**W**) and azithromycin (**W**) The most common diagnoses for antibiotics were URTIs and sinusitis
Manderson 2020 [59]. Interviews with patients or guardians	Qualitative study involving observing patient/provider consultations and 65 in-depth interviews with patients, parents or guardians presenting with children among CHCs and private GPs across South Africa Objective was to explore providers’ and patients’ expectations for treating ARIs with antibiotics to provide future guidance	Prescribers’ treatment decisions were typically informed by a clinical assessment, concern about the risks of bacterial infection and their perceptions of the patient’s ability to seek further care Physicians in private practice often provided a prescription—sometimes post-dated to discourage antibiotic use—but with the option for antibiotics if needed Patients did not always request antibiotics and were willing to discuss how to manage acute conditions without them, which is encouraging for the future
Boffa et al., 2021 [99]; Salomon et al., 2022 [100]. Standardised patients	Assessment of antibiotic prescribing as inappropriate prescribing can mask TB symptoms as well as lead to TB diagnostic delay A total of 511 standardised patients (SPs) visited 212 private GPs in Cape Town and Durban Three SP scenarios: (i) TB symptoms, HIV-positive; (ii) TB symptoms coupled with a positive molecular test for TB and HIV-negative; (iii) TB symptoms, history of incomplete TB and treatment, HIV-positive	Antibiotics were prescribed on 76.5% of occasions (95% CI 71.7% to 80.7%), making them the most common medicines prescribed 86.1% of antibiotics prescribed belonged to the ‘Access’ group (with amoxicillin and co-amoxiclav accounting for 64%); fluoroquinolones (**W**) accounted for 8.8% (95% CI 6.3% to 12.3%); none in the ‘Reserve’ group Factors associated with antibiotic prescribing included whether the SP was asked to follow up if symptoms persisted and if the SP presented as HIV-positive An injection was offered in 31.9% of visits—92% with unexplained rationale 61.8% of medicines prescribed were not listed in the South African Primary Healthcare EML Overall, there were concerns about inappropriate antibiotic prescribing and costs
Alabi et al., 2022 [46]. Retrospective analysis of a claims database of a health insurer	Assessment of dosing and duration of prescribed antibiotic therapy Analysis of 188,141 antibiotic prescriptions for 174,889 patients, including their appropriateness based on ICD-10 classification	*Diagnoses (Principal)*: Diseases of the respiratory system (J00–J99): 46.1% of all diagnoses Unknown diagnosis: 84.0% of Z00–Z99, i.e., factors influencing health status and contact with health services (accounting for 15.8% of all diagnoses) Diseases of genitourinary system (N): 8.92% *Classes of antibiotics prescribed (Principal)*: Penicillins were the most frequently prescribed antibiotic class (40.7% of all antibiotics prescribed) Macrolides (16.8%), cephalosporins (15.7%) and quinolones (13.1%) were also frequently prescribed *Individual antibiotics prescribed (Principal)*: Co-amoxiclav (**A**): 28.6% of all antibiotics prescribed Amoxicillin (**A**): 9.8%; azithromycin (**W**): 9.3% Ciprofloxacin (**W**): 8.5%; cefpodoxime (**W**): 6.7% *Appropriateness of the prescriptions*: 8.8% of all the prescriptions were appropriate 32.0% of all prescriptions were potentially appropriate, 45.4% inappropriate and 13.8% could not be assessed due to a lack of specific code, containing unlisted codes or having unclear descriptions *Appropriately and potentially appropriately prescribed antibiotics*: 57.7% of antibiotic prescriptions were prescribed at the correct doses 27.4% of antibiotics were prescribed with wrong doses 14.9% of prescriptions could not be assessed
Guma et al., 2022 [33]. Study among private GPs using a semi-structured web-based questionnaire	Assess current empiric prescribing of antibiotics among private GPs for patients with ARIs and associated key factors A semi-structured web-based questionnaire was used to document the findings among 209 GPs	55.5% of surveyed private GPs prescribed antibiotics empirically for patients with ARIs more than 70% of the time The prescribing of antibiotics was primarily for symptom relief and the prevention of complications GPs between the ages of 35 and 44 years or >55 years and in practice for < 15 years were significantly more likely to prescribe antibiotics empirically Three key factors were significantly associated with empiric prescribing: workload/time pressures, diagnostic uncertainty and the use of a formulary GPs with more experience and working alone were slightly less likely to prescribe antibiotics empirically
Lagarde and Blaauw, 2019 and 2023 [38,62,90]. Simulated patients	The objective was to assess prescribing practices for young and healthy simulated patients (SP) presenting with viral bronchitis in the city of Johannesburg A total of 99 SP visits in the private sector (and 102 in the public sector) One hundred twenty-five providers (across sectors) were interviewed face-to-face	Antibiotics were recommended in 72.6% of consultations, lower in the private sector (66.7%) versus the public sector (78.4%) The high rates of antibiotic prescribing were not helped by the following: ○ Significant knowledge gaps, particularly in relation to the recommended management of acute bronchitis ○ Perceived patient pressure to prescribe antibiotics for this condition 30% of patients prescribed antibiotics received amoxicillin (**A**), 35% co-amoxiclav (**A**) and 14% clarithromycin (**W**) Overall, in 20% of consultations in the private sector, antibiotics from the ‘Watch’ list were prescribed (versus only 5% in the public sector) 84% of providers knew that the SP case was likely a viral infection (88% in the private sector vs. 77% in the public sector) 58% of providers believed antibiotics would not hasten recovery (68% in the private sector vs. 40% in the public sector; *p* = 0.002) 72% of private providers believed patients would not come back if no antibiotic were prescribed (72% vs. 47% in the public sector; *p* = 0.008) Consultations that occurred later in the day were associated with greater prescribing of antibiotics—possibly due to fatigue Higher awareness of AMR among private vs. public providers (*p* < 0.001)

NB{ AMR: Antimicrobial Resistance; ARI: Acute Respiratory Infections; * AWaRe classification: Access (**A**) and Watch (**W**) antibiotics [34,93,94]; CHCs: Community Health Centres; EML: Essential Medicine List; GP: General Practitioner; SP: Simulated patients; TB: Tuberculosis; URTI: Upper Respiratory Tract Infection.

**Table 6 antibiotics-12-01540-t006:** Antimicrobial stewardship programmes that have been implemented in ambulatory care in South Africa and their impact.

Author and Year	Setting and Activities	Key Findings Including Impact
Blaauw and Lagarde, 2019 [38]	Study using mystery patients being treated either by nurses in PHCs or by Private GPs for acute bronchitis In a follow-up study, mystery patients explicitly told GPs and nurses that they did not want antibiotics unless they were really necessary	Antibiotic prescribing for acute bronchitis decreased by 20% compared with the first study (Table 1 and Table 2) Despite this, more than half of patients still received antibiotics even after stating they did not want one
van Hecke et al., (2019), and Epps et al., 2021 [69,106]	Explore the perceptions among 23 HCPs in publicly funded PHCs Using semi-structured questionnaires to document attitudes and experiences of existing POCTs as well as barriers and opportunities to introducing (hypothetical) new POCTs	Largely positive experiences among HCPs with currently available POCTs HCPs were optimistic about the potential for new POCTs to support evidence-based prescribing decisions to reduce unnecessary antibiotic prescribing and to reduce the need for further investigations. In addition, support effective communication with patients, especially when antibiotics are unlikely to be beneficial However, resources, available space and workflow disruption are currently seen as barriers to their uptake into routine care
De Vries et al., 2022 [89]	Multidisciplinary audit and feedback meetings once a month at 13 PHCs Ten antibiotic prescriptions were randomly selected for a peer review audit by the team, assessed and scored for adherence to seven key measures, including antibiotic choice according to the STG/EML All measures had to be met for the prescription to be considered correct Concurrently, primary care pharmacists monitored monthly consumption for the six oral antibiotics most prescribed, e.g., amoxicillin (**A**), co-amoxiclav (**A**), penicillin (**A**), azithromycin (**W**), ciprofloxacin (**W**) and flucloxacillin (**A**)—DDDs/100 prescriptions dispensed	Mean overall level of adherence to guidelines increased from 11% in July 2017 to 53% in June 2019 However, prescribing adherence was significantly lower in the winter and spring, concurrent with higher antibiotic prescribing and consumption—this may reflect inappropriate antibiotic prescribing for increased viral ARIs during these months Mean of 19% correct prescriptions in the first 6 months (baseline) to a mean of 47% correct prescriptions in the last 6 months of the study (*p* < 0.001) Reduction of 12.9 DDDs between the pre- and post-intervention periods (*p* = 0.0084) was documented, i.e., a 19.3% decrease in antibiotic consumption
Masetla et al., 2023 [107]	The aim of the study was to provide AMS services to patients in a hospital’s outpatient department with chronic bone and joint infections presenting to the orthopaedic clinic Forty-four patients participated, with questionnaires used to assess the understanding of their conditions as well as adherence to prescribed antibiotics Review of antibiotic prescriptions with prescribers contacted if concerns including adherence to current STGs/EML	Seventy-one antibiotics were prescribed, with 62% from the ‘Watch’ group A total of 239 interventions were made, including educating patients and clinicians The majority of interventions regarding patients were concerning knowledge of their condition and medication (n = 145; 61%) Sixty-five interventions (27%) were made regarding educating patients on adherence to prescribed antibiotics and their importance in helping resolve their condition The majority (96%) of the antibiotics were not prescribed according to the STG; however, interventions were only needed in 31% of prescribed antibiotics (n = 71) since the STG only recommends empiric therapy directed against *Staphylococcus aureus* The majority of the drug treatment interventions (n = 29) for the appropriate antibiotic selection (62%)

NB: AMS: Antimicrobial Stewardship; ARI: Acute Respiratory Illness; AWaRe classification for Access (**A**) and Watch (**W**) antibiotics [93,94]; DDD: defined daily dose; GP: General Practitioner; HCP: Healthcare Professional; PHC: Primary Healthcare Clinic; POCT: Point-of-Care Testing; URTI: upper respiratory tract infection; STGs/EML: Standard Treatment Guidelines/Essential Medicine List.

**Table 7 antibiotics-12-01540-t007:** Suggested activities in the short to medium term to reduce inappropriate prescribing of antibiotics in primary care settings in South Africa.

Activity
National and regional health authorities and private insurance companies must regularly monitor antibiotic prescribing habits of HCPs in primary care as part of the NAP, given current concerns. The routine instigation of EHRs/easy-to-use electronic applications (Apps) is essential going forward to facilitate audit and feedback activities
National and regional health authorities and private insurance companies must work closely with HCPs to agree on future prescribing and quality indicators. Existing indicators can be used as a starting point (Table 5)
Potentially update current medicine list and guidelines, e.g., South African EML/STGs (2020 Edition), based on the newly published AWaRe guidance where pertinent
Seek to instigate pertinent ASPs in primary healthcare across the sectors based on agreed-upon quality indicators and prescribing guidance.
Regularly monitor adherence to guidelines as part of ASPs through feedback/audit activities that are in line with the goals of the NAP. As part of this, ensure HCPs are fully aware of pertinent diagnostic codes
Encourage multidisciplinary collaboration to improve future antibiotic prescribing across the sectors
Improve the education of patients to ensure they are familiar with terms such as ASPs. In addition, antibiotics are not appropriate for viral infections and will not alter the disease process; however, such activities will increase AMR and adverse events
Longer term—encourage more citizens to become antibiotic guardians
Refining curricula among student healthcare professionals to improve their knowledge regarding antibiotics, AMR and ASPs, as well as continuous professional development activities post qualification to address knowledge and training gaps

NB: AMR: Antimicrobial Resistance; ASPs: Antimicrobial Stewardship Programmes; AWaRe: Access, Watch, Reserve [93,94]; EHR: Electronic Healthcare Records; HCPs: Healthcare Professionals; NAP: National Action Plan; STGs/EML: Standard Treatment Guidelines/Essential Medicine List.

## Data Availability

We have already referenced all sourced papers and publications.

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
