# Peer review of "A Narrative Review of Antibiotic Prescribing Practices in Primary Care Settings in South Africa and Potential Ways Forward to Reduce Antimicrobial Resistance"

_antibiotics, 2023, doi:10.3390/antibiotics12101540_

Round 1
Reviewer 1 Report
Overall: The submitted work is a narrative review focused on antibiotic prescribing practices in primary care settings in South Africa and its association with rising antimicrobial resistance (AMR) rates. The review sheds light on notable antibiotic utilization patterns and activities to improve prescribing. The authors also emphasize the significance of prescribing antibiotics from the ‘Access’ list as opposed to the ‘Watch’ list. There is a critical call for the enhancement of education among all relevant groups and the importance of consistent monitoring with established guidelines and indicators. However, the review could benefit from some minor revisions.
It would be advantageous for the authors to integrate dissemination and implementation science into the discussion section. This could help guide strategies to promote the uptake and application of proven intervention strategies (e.g. peer to peer comparison, commitment posters) in real-world settings.
Title: The title should specify that this is a "Narrative Review" to ensure clarity for readers right from the onset.
Abstract: The flow in the abstract would benefit from clearer delineation between background information and the data derived from the literature. Explicitly mentioning that the review summarizes findings from various studies will help orient the reader. The reference to the ‘Access’ list and the ‘Watch’ list can be clarified by mentioning the WHO AWaRe list for better context.
Author Response
Quality of English Language
( ) I am not qualified to assess the quality of English in this paper
( ) English very difficult to understand/incomprehensible
( ) Extensive editing of English language required
( ) Moderate editing of English language required
( ) Minor editing of English language required
(x) English language fine. No issues detected
Author comments: Thank you – appreciated
Yes |
Can be improved |
Must be improved |
Not applicable |
|
Does the introduction provide sufficient background and include all relevant references? |
(x) |
( ) |
( ) |
( ) |
Are all the cited references relevant to the research? |
(x) |
( ) |
( ) |
( ) |
Is the research design appropriate? |
(x) |
( ) |
( ) |
( ) |
Are the methods adequately described? |
(x) |
( ) |
( ) |
( ) |
Are the results clearly presented? |
(x) |
( ) |
( ) |
( ) |
Are the conclusions supported by the results? |
(x) |
( ) |
( ) |
( ) |
Comments and Suggestions for Authors
1) Overall: The submitted work is a narrative review focused on antibiotic prescribing practices in primary care settings in South Africa and its association with rising antimicrobial resistance (AMR) rates. The review sheds light on notable antibiotic utilization patterns and activities to improve prescribing.
Author comments: Thank you for this summary – appreciated!
2) The authors also emphasize the significance of prescribing antibiotics from the ‘Access’ list as opposed to the ‘Watch’ list. There is a critical call for the enhancement of education among all relevant groups and the importance of consistent monitoring with established guidelines and indicators.
Author comments: Thank you again – we also appreciate this!
3) However, the review could benefit from some minor revisions. It would be advantageous for the authors to integrate dissemination and implementation science into the discussion section. This could help guide strategies to promote the uptake and application of proven intervention strategies (e.g. peer to peer comparison, commitment posters) in real-world settings.
Author comments: Thank you – We have now added in more details into the Discussion that multiple activities are typically needed to enhance future prescribing habits. This includes details regarding the ‘Wise List’ in Stockholm, Sweden, and examples with Biosimilars. We trust this is now acceptable.
4) Title: The title should specify that this is a "Narrative Review" to ensure clarity for readers right from the onset.
Author comments: Thank you – now done
5) Abstract: The flow in the abstract would benefit from clearer delineation between background information and the data derived from the literature. Explicitly mentioning that the review summarizes findings from various studies will help orient the reader. The reference to the ‘Access’ list and the ‘Watch’ list can be clarified by mentioning the WHO AWaRe list for better context.
Author comments: Thank you – now updated. We trust this is now acceptable.

Reviewer 2 Report
The manuscript consists of 29 pages in total, including 5 tables and the list of total 159 literature references. The article presents the results of the study on improving antibiotic rational use in practice in South Africa. As such, the article is meant as rising current and highly practice-relevant problem and thus it may potentially be interesting for the Readers, fitting in the scope of works published in the Journal. The Title of the article is relevant to its contents and informative enough.
The Abstract is unclear, the Authors introduce terms "Access/Watch list" or "key stakeholder groups" without any explanation.
All abbreviations, no matter how seemingly obvious, shall be explained while used for the first time in the text, separately in the Abstract and separately in the main text of the article, e.c. line 49.
The Authors shall improve their way to deal with literature references, as they notoriously refer at once to several literature sources, most of them used only once through the text, when supporting simple statements, while each literature reference shall provide its unique information input to the text. The current practice results in horrifically expanded number of literature references on the respective list, which actually do not add anything to the merit value of the text.
The English language quality is acceptable, however the style of the text is heavy and exhausting in reception, with long and convoluted sentences, sometimes seemingly missing verbs, see e.c. lines 31-33.
Also mixing the two styles of referring to literature - [numbers in brackets] versus authors' names with year of publication in (brackets) - is likely to annoying the Readers as it seriously blurs the content of communicates in the text; the Authors shall just stick just to [numbers in brackets] style and remove from the text the excessive names and years, as it does not add anything to the merit of the main text of the article who actually published the data as long as all the used sources are treated as trusted ones, as they by definition shall be.
I do not see any added value to a narrative review in placing the data in tables 1-5 that just cite the findings from the literature. In my opinion, the Authors shall stick to the convention of continuous text that contains - logically arranged selected into a single coherent line of argumentation - valuable information from used literature sources rather than cite it directly, which unnecessarily expands the words volume of the article.
Also I do not see any reason why the text in table 6 shall be aligned to the center, which makes it much more difficult to read; after all, such a massive amount of continuous text shall not be put into a table flowing through several pages, leaving headers behind on previous pages, but rather converted into a numbered list. The "suggested activities" shall be concisely stated and highly concrete Authors' choice of suggested solutions - while now there are multiple general sentences written in narrative or even discussion style in each point, some with literature references and most without, which does not serve the declared aim adequately.
In general, the article is very difficult to read, the line of argumentation seems unfocused and difficult to follow, in particular it does not, in a clear enough way, lead to the Conclusions, which are not concrete enough for an article that is meant as a sort of guide in solving a problem.
The abundant literature references list lacks DOI links to sources, which would significantly add to the value of the manuscript as a potentially educational source.
The English language quality is acceptable, however the style of the text is heavy and exhausting in reception, with long and convoluted sentences, which negatively adds to mixing the two styles of referring to literature, leading to heavy excess of words in sentences in comparison to the intended communicates.
Author Response
Quality of English Language
( ) I am not qualified to assess the quality of English in this paper
( ) English very difficult to understand/incomprehensible
( ) Extensive editing of English language required
(x) Moderate editing of English language required
( ) Minor editing of English language required
( ) English language fine. No issues detected
Author comments: Thank you – we have revised the paper where we can to address this with the help of some of the co-authors who are native English speakers (one of whom has over 500 publications in peer-reviewed Journals since 2008) and hope this is now OK.
Yes |
Can be improved |
Must be improved |
Not applicable |
|
Does the introduction provide sufficient background and include all relevant references? |
(x) |
( ) |
( ) |
( ) |
Are all the cited references relevant to the research? |
( ) |
(x) |
( ) |
( ) |
Is the research design appropriate? |
( ) |
(x) |
( ) |
( ) |
Are the methods adequately described? |
( ) |
(x) |
( ) |
( ) |
Are the results clearly presented? |
( ) |
( ) |
(x) |
( ) |
Are the conclusions supported by the results? |
( ) |
(x) |
( ) |
( ) |
Comments and Suggestions for Authors
1) The manuscript consists of 29 pages in total, including 5 tables and the list of total 159 literature references. The article presents the results of the study on improving antibiotic rational use in practice in South Africa. As such, the article is meant as rising current and highly practice-relevant problem and thus it may potentially be interesting for the Readers, fitting in the scope of works published in the Journal. The Title of the article is relevant to its contents and informative enough.
Author comments: Thank you for these positive comments – appreciated!
2) The Abstract is unclear, the Authors introduce terms "Access/Watch list" or "key stakeholder groups" without any explanation.
Author comments: Thank you – now updated. We trust this is now acceptable.
3) All abbreviations, no matter how seemingly obvious, shall be explained while used for the first time in the text, separately in the Abstract and separately in the main text of the article, e.c. line 49.
Author comments: Thank you – now updated.
4) The Authors shall improve their way to deal with literature references, as they notoriously refer at once to several literature sources, most of them used only once through the text, when supporting simple statements, while each literature reference shall provide its unique information input to the text. The current practice results in horrifically expanded number of literature references on the respective list, which actually do not add anything to the merit value of the text.
Author comments: Thank you for this comment. We have looked at the references and seen where we cut these down – especially for non-South African references. However, this has proved difficult especially following requests from other Reviewers. We must admit this is the first time we have experienced comments such as this. Typically, academic reviewers/ journals wish multiple supporting references – especially where there are concerns/ challenging issues. This is certainly the experience of one of the corresponding authors with over 500 publications in peer-reviewed Journals since 2008 – many of which are in good impact Journals. We trust this is OK with you.
5) The English language quality is acceptable, however the style of the text is heavy and exhausting in reception, with long and convoluted sentences, sometimes seemingly missing verbs, see e.c. lines 31-33.
Author comments: Thank you – we have revised the paper where we can to address this with the help of some of the co-authors who are native English speakers (one of whom has over 500 publications in peer-reviewed Journals since 2008) and hope this is now OK.
6) Also mixing the two styles of referring to literature - [numbers in brackets] versus authors' names with year of publication in (brackets) - is likely to annoying the Readers as it seriously blurs the content of communicates in the text; the Authors shall just stick just to [numbers in brackets] style and remove from the text the excessive names and years, as it does not add anything to the merit of the main text of the article who actually published the data as long as all the used sources are treated as trusted ones, as they by definition shall be.
Author comments: Thank you for this comment. However – you are probably unfamiliar with the reference style of the Journal which asks for references in [brackets] as we have done. In addition – in the experience of the co-authors (as mentioned one of whom has published over 500 papers in peer-reviewed Journals since 2008) – typically when naming an author followed by et al – the accepted academic etiquette is to insert the year of publication after ‘et al’ as we have done. We trust this helps explain our approach and hope this is now OK.
7) I do not see any added value to a narrative review in placing the data in tables 1-5 that just cite the findings from the literature. In my opinion, the Authors shall stick to the convention of continuous text that contains - logically arranged selected into a single coherent line of argumentation - valuable information from used literature sources rather than cite it directly, which unnecessarily expands the words volume of the article.
Author comments: We are sorry – we just do not understand this comment. What we have tried in the Tables to summarise key aspects of relevant papers for academic readers, government/ health authority personnel as well as interested healthcare professionals and patients as time is typically scare for many of these key groups. This is in line with many publications we have co-authored summarising key points with such personnel in the past. Subsequently, use the text to summarise the key findings from the Tables – including key findings on prescribing habits/ differences between the sectors, etc., with the Tables including additional details for the readers if needed. In this way – the typical reader can quickly go over the text and refer back to key elements in the Tables where pertinent. We have also added in a further Table (Table 1) following comments from one of the Reviewers to summarise ongoing Government initiatives, etc., in South Africa to enhance appropriate prescribing of antibiotics across the sectors. We trust this is now OK.
8) Also I do not see any reason why the text in table 6 shall be aligned to the center, which makes it much more difficult to read; after all, such a massive amount of continuous text shall not be put into a table flowing through several pages, leaving headers behind on previous pages, but rather converted into a numbered list. The "suggested activities" shall be concisely stated and highly concrete Authors' choice of suggested solutions - while now there are multiple general sentences written in narrative or even discussion style in each point, some with literature references and most without, which does not serve the declared aim adequately.
Author comments: Thank you for this. We have now consolidated possible activities in a revised Table (Table 7) with expansion in Supplementary Table S1 for those who are interested. The same applies to other examples of ASPs in ambulatory care in LMICs (Table S2). The formatting of the Table will be addressed if and when the paper is accepted and we receive the first proof from the Journal. We trust this is OK with you.
9) In general, the article is very difficult to read, the line of argumentation seems unfocused and difficult to follow, in particular it does not, in a clear enough way, lead to the Conclusions, which are not concrete enough for an article that is meant as a sort of guide in solving a problem.
Author comments: Thank you for this. We beg to differ as we have tried to lay out quite clearly in 4.1 – and summarised at the start of the Results section - why we have adopted this logical approach. The principal Tables for all key stakeholders are Tables 7 and S1 as we believe this clearly lays out for all key stakeholder groups (based on the considerable experience of the co-authors) what are principal ways forward/ suggested activities to improve future prescribing of antibiotics in ambulatory care in South Africa to reduce AMR. In this way – serve as a conclusion and next steps. We have now updated the conclusion, and trust everything is now acceptable.
10) The abundant literature references list lacks DOI links to sources, which would significantly add to the value of the manuscript as a potentially educational source.
Author comments: Thank you – this will be sorted out with the Journal when and if the paper is accepted for publication.
11) Comments on the Quality of English Language - The English language quality is acceptable, however the style of the text is heavy and exhausting in reception, with long and convoluted sentences, which negatively adds to mixing the two styles of referring to literature, leading to heavy excess of words in sentences in comparison to the intended communicates.
Author comments: Thank you – we have revised the paper where we can to address this with the help of some of the co-authors who are native English speakers (one of whom has over 500 publications in peer-reviewed Journals since 2008) and hope this is now OK.

Reviewer 3 Report
The authors have conducted a comprehensive review regarding antibiotic prescribing practices in South Africa. The topis is of utmost interest in the current era of AMR and this is a very well planned review giving a broad view covering numerous aspects.
The manuscript is quite well written. The methodology is precise and reproducible. The results are well depicted and discussed. Few soft suggestions from my side are:
· Evidence on antibiotic prescribing practices in other LMICs in general may be added in introduction.
· What proportion of patients get treatment from pubic versus private sector in the country?
· A review of any Quality improvement studies or studies assessing impact of educational interventions pertaining to antibiotic prescribing or AMR per se conducted across the country may be added.
· For the prescribing and quality indicators used in South Africa, do they not include number (percentage) of prescriptions from WHO/ National essential medicine list; and proportion of generic prescriptions as are being included in WHO core prescribing indicators?
· Any such extensive review conducted in other parts of the world may be added and compared in discussion.
Author Response
Quality of English Language
( ) I am not qualified to assess the quality of English in this paper
( ) English very difficult to understand/incomprehensible
( ) Extensive editing of English language required
( ) Moderate editing of English language required
( ) Minor editing of English language required
(x) English language fine. No issues detected
Author comments: Thank you – appreciated.
Yes |
Can be improved |
Must be improved |
Not applicable |
|
Does the introduction provide sufficient background and include all relevant references? |
(x) |
( ) |
( ) |
( ) |
Are all the cited references relevant to the research? |
(x) |
( ) |
( ) |
( ) |
Is the research design appropriate? |
(x) |
( ) |
( ) |
( ) |
Are the methods adequately described? |
(x) |
( ) |
( ) |
( ) |
Are the results clearly presented? |
(x) |
( ) |
( ) |
( ) |
Are the conclusions supported by the results? |
(x) |
( ) |
( ) |
( ) |
Comments and Suggestions for Authors
1) The authors have conducted a comprehensive review regarding antibiotic prescribing practices in South Africa. The topic is of utmost interest in the current era of AMR and this is a very well planned review giving a broad view covering numerous aspects.
Author comments: Thank you for these kind words – appreciated!
2) The manuscript is quite well written. The methodology is precise and reproducible. The results are well depicted and discussed.
Author comments: Thank you for these kind words – appreciated!
3) Few soft suggestions from my side are:
- a) Evidence on antibiotic prescribing practices in other LMICs in general may be added in introduction
Author comments: Thank you for this. We have now enhanced the introduction to include comments from other LMICs particularly African countries, and trust this is now acceptable.
- b) What proportion of patients get treatment from pubic versus private sector in the country?
Author comments: Thank you – we have now added in these details.
- c) A review of any Quality improvement studies or studies assessing impact of educational interventions pertaining to antibiotic prescribing or AMR per se conducted across the country may be added
Author comments: Thank you for this comment. We have now added in a new Table (Table 1) to document current quality improvement activities regarding antibiotics in ambulatory care in South Africa. Table 5 contains details of quality/ prescribing indicators that have been used to date in South Africa with Table 6 giving details of ASPs to date and their outcome in South Africa. We trust this is now OK.
- d) For the prescribing and quality indicators used in South Africa, do they not include number (percentage) of prescriptions from WHO/ National essential medicine list; and proportion of generic prescriptions as are being included in WHO core prescribing indicators?
Author comments: Thank you. Table 5 includes indicators that have been used to date. It is likely that new indicators will be developed around the AWaRe book to give a better idea of the quality/ appropriateness of antibiotic prescribing than just documenting the number of prescriptions which does not really give any information regarding the quality of prescribing (which some of the co-authors have published on). Similarly for ‘generic’ prescriptions – this is not really an issue in the public healthcare system in South Africa since all procurement for healthcare facilities is based on INNs. Physicians in the private system are also encouraged to prescribe the generic when available once the originator has lost its patent – although prices vary by Health Insurance Group (which we have also published on). We trust this is acceptable.
- e) Any such extensive review conducted in other parts of the world may be added and compared in discussion.
Author comments: Thank you – we have added details of other countries, including other reviews, especially other Africa countries where we can. However – based on our extensive experience we believe this is the first study to really pull all these strands together. We hope this is OK.

Round 2
Reviewer 2 Report
The manuscript has been significantly improved and at places also expanded in comparison to its previous version - in particular, it is now more narrative and thus easier to read.
I would advise to remove the references to resources (tables) from the Conclusions section and move them into the Discussion section.
I have no further suggestions of improvements.